# The Double Game Played by Th17 Cells in Infection: Host Defense and Immunopathology

**DOI:** 10.3390/pathogens11121547

**Published:** 2022-12-15

**Authors:** Marino Paroli, Rosalba Caccavale, Maria Teresa Fiorillo, Luca Spadea, Stefano Gumina, Vittorio Candela, Maria Pia Paroli

**Affiliations:** 1Division of Clinical Immunology, Department of Clinical, Anesthesiologic and Cardiovascular Sciences, Sapienza University of Rome, 00185 Rome, Italy; 2Department of Biology and Biotechnology “Charles Darwin”, Sapienza University of Rome, 00185 Rome, Italy; 3Post Graduate School of Public Health, University of Siena, 53100 Siena, Italy; 4Department of Anatomy, Histology, Legal Medicine and Orthopedics, Sapienza University of Rome, 00185 Rome, Italy; 5Eye Clinic, Department of Sense Organs, Sapienza University of Rome, 00185 Rome, Italy

**Keywords:** Th17, pathogens, immunopathology

## Abstract

T-helper 17 (Th17) cells represent a subpopulation of CD4+ T lymphocytes that play an essential role in defense against pathogens. Th17 cells are distinguished from Th1 and Th2 cells by their ability to produce members of the interleukin-17 (IL-17) family, namely IL-17A and IL-17F. IL-17 in turn induces several target cells to synthesize and release cytokines, chemokines, and metalloproteinases, thereby amplifying the inflammatory cascade. Th17 cells reside predominantly in the lamina propria of the mucosa. Their main physiological function is to maintain the integrity of the mucosal barrier against the aggression of infectious agents. However, in an appropriate inflammatory microenvironment, Th17 cells can transform into immunopathogenic cells, giving rise to inflammatory and autoimmune diseases. This review aims to analyze the complex mechanisms through which the interaction between Th17 and pathogens can be on the one hand favorable to the host by protecting it from infectious agents, and on the other hand harmful, potentially generating autoimmune reactions and tissue damage.

## 1. Introduction

T helper 17 cells represent a subset of CD4+ T lymphocytes discovered about 20 years ago [1,2]. It was evident from the beginning that this population played a key role in defense against bacteria, viruses, and fungi. It was shown that the defensive action of Th17 cells occurred primarily through the production of members of the IL-17 family, in particular IL-17A and IL-17F [3,4]. It was also found that the specific cytokine signature of Th17 cells was closely dependent on IL-23 produced by cells of innate immunity in response to stimulation by microbial agents. The dependence of IL-17 production on IL-23 led to the formulation of the term “IL-23/IL-17 axis” [5].

Although Th17 cells were shown to be essential in preserving the integrity of the mucosal barrier from attack by infectious agents, subsequent studies revealed that these cells were also involved in immunopathological processes. Indeed, in an appropriate inflammatory microenvironment, they were found to be responsible for immunopathology, promoting the induction and maintenance of inflammatory and autoimmune diseases [6,7,8,9,10].

The purpose of this review is therefore to summarize the discoveries that led to the identification of Th17 cells and to analyze the complex mechanisms of their differentiation and function, with a focus on how their interaction with pathogens can lead on the one hand to defense against the infectious agents and on the other hand to the generation of immunopathology, as emerged from experimental animal models and human study.

## 2. Th17 Cell Discovery: A Novel CD4+ T-Helper Cell Paradigm

For several years since the second half of the 1980s, CD4+ T cells have been subdivided into two subpopulations, T-helper 1 (Th1) cells and Th2 cells [11]. Th1 cells were characterized by their ability to produce interferon (IFN)-γ, cooperate with B cells to produce immunoglobulin (Ig)G, and give rise to delayed-type T-cell responses [12]. Conversely, Th2 cells were characterized by the ability to produce interleukin (IL)-4, which is required to promote isotope switching by B cells for IgE production [13,14]. While Th1 cells predominantly provided defense against intracellular pathogens, Th2 cells provided defense against parasites. In 1989, the so-called Th1/Th2 paradigm was thus formulated [15]. 

Subsequently, however, several pieces of evidence showed that IFN-γ production by Th1 cells did not fully explain the presence of pro-inflammatory CD4+ T-cell mediated responses in mouse models [16,17,18]. The Th1/Th2 paradigm was therefore destined to be overcome. 

## 3. IL-17-Member Family

A critical event that preceded the discovery of Th17 cells was the cloning of a new interleukin with pro-inflammatory properties in 1993 [19,20]. This interleukin was initially defined as CTLA8 and later as interleukin-17A. That interleukin showed sequence homology with an open reading frame of the *Herpesvirus saimiri*, a herpes virus capable of infecting T cells [21,22]. Interleukin 17A was characterized by not having sequence homology with other known cytokines and deserved the definition of cytokine because it could induce the production of immune-active soluble factors by target cells [23]. Other cytokines structurally similar to interleukin 17A were then identified in sequence homology studies [24,25,26]. All these molecules were then grouped into a family defined as IL-17 that presently includes six members from 17A to IL-17F [27,28]. Five specific receptors expressed on different cell types were then cloned [29,30,31,32]. These receptors have molecular peculiarities that differentiate them from other interleukin receptors. In particular, they contain an intracytoplasmic motif termed SEFIR (SEF/IL-17 receptor) that has some similarities with a domain present in the Toll/IL-1 receptor (TIR) [26]. IL-17R signaling initiates with the recruitment of Act1 adaptor molecule, with subsequent IL-17R/Act-1 association [33,34,35] thus amplifying the signal transduction. This interaction is critical in the response to pathogens as demonstrated in murine models of knockout mice for the gene encoding IL-17RA as well as in humans with *ACT1* mutations or with congenital deficiency of IL-17A or IL-17C, where a high susceptibility to fungal infections is observed [36]. Act1 possesses the peculiar ability to bind to E3 Ubiquitin [37]. Through this interaction, TNF-receptor associated factor(TRAF)6 is then recruited [34,38], resulting in activation of the nuclear factor kappa-light-chain-enhancer of activated B (NF-κB) and subsequent gene transcription of several antimicrobial proteins [20,38,39,40,41]. 

In the year 2000, a discovery would prove essential for the subsequent identification of new pro-inflammatory CD4+ T cells in addition to the classic Th1 and Th2 cells [42]. Indeed, a new cytokine called p19 was discovered. This cytokine was able to form a heterodimer with the p40 chain of IL-12. The association of these two proteins originated a new interleukin called IL-23. Interleukin-23 was able to bind a receptor to IL-23R constituted by IL23R/IL-12β1 heterodimer [43]. Later studies showed that IL-23 was produced mainly by dendritic cells after activation by prostaglandin E and adenosine triphosphate [44,45]. It was shown that IL-23 was able to induce IL-17 production by a subpopulation of CD4+ T cells distinct from both Th1 and Th2 cells [46,47]. Therefore, this subpopulation was termed Th17 [1,2]. The peculiar differentiation and function characteristics of this subpopulation were further defined [7]. Th17 cells predominantly produce a member of the IL-17 family, namely IL-17A. However, subsequent study showed that other soluble factors involved in the inflammatory response, such as IL-17F, IL-21, IL-22, IL-26, and the chemokines CXCL8 and CCL20, were also produced [5]. It was found that a peculiar feature of Th17 cells was that their differentiation depended on specific transcription factors retinoic-acid orphan gamma receptor t (RORγt) in mice and its human isoform retinoic-acid orphan receptor C (RORC) [48,49]. These factors in turn induced *il17a* gene transcription [6,50]. Th17 cells were shown to characteristically express CCR6 chemokine receptor on their surface [4]. Further studies identified the existence of an intermediate cell subtype termed Th1Th17 able to produce both IL-17 and IFN-γ [3,51,52].

## 4. Differentiation of Th17 Cells

Differentiation of Th17 cells is a rather intricate molecular process that has been clarified at least in part only recently. This process requires TCR engagement of naïve CD4+ T cells together with the action of different cytokines present in the microenvironment. In more detail, IL-6 and tissue growth factor-β (TGF-β) provide to chromatin remodeling of the *il17* gene locus [17,53,54]. IL-6 enhances retinoic-acid orphan receptors (RORγt) transcription through signal transducer and activator of transcription 3 (STAT3) phosphorylation [17], whereas TGF-β regulates Th17 differentiation via Staufen1 (STAU1)-mediated mRNA decay (SMD)-dependent or -independent pathways. It is noteworthy that TGF-β is also capable to promote regulatory T-cell (Treg) differentiation, which in turn suppresses Th17 through the function of forkhead box P3 (FOXP3). However, this activity is counteracted by IL-6-phosphorylated STAT3, which downregulates FOXP3, with consequent induction of RORγt and transformation of inducible Tregs into Th17 cells [50]. An important role in the early stages of Th17 cell differentiation is also played by IL-1β. This interleukin upregulates the expression of interferon regulatory factor (IRF) 4 [55] and RORγt [56,57,58]. Once differentiated, Th17 cells express IL23R on their surface. Interaction with IL-23 present in the inflammatory microenvironment is required to maintain the phenotype of Th17 cells, increasing RORγt and IL-17 expression by the intervention of STAT3 [59], but does not participate in the Th17 differentiation process [8,60]. Figure 1 shows the differentiation process of Th17 cells.

## 5. Plasticity of Th17

An important feature of Th17 cells is their plasticity, defined by their ability to acquire functional and phenotypic characteristics of other CD4+ T lymphocyte subpopulations. Indeed, Th-17 cells can acquire the characteristics of Th1 cells in the presence of interleukin 12, which downregulates the expression of RORγt/RORC and induces the expression of T-bet, a major transcription factor of IFN-γ [61,62,63]. In a microenvironment rich in IL-4, they can acquire a Th2 phenotype [64]. Th17 cells present in the Peyer’s patches can acquire the follicular T cells (Tfh) phenotype and induce immunoglobulin (Ig) A production by geminal center B lymphocytes [65]. Finally, differentiation of naïve CD4+ T cells into Treg or Th17 is bidirectional. Interconnection between these two subpopulations depends on FOXP3/ RORγt/RORC balance, as demonstrated in numerous studies. This is regulated by the relative activity of several cytokines, including TGF-β, IL-6, IL-21, IL1β, and IL-23 [66,67,68]. Th17 cells exert their function on several cellular targets that are not only part of the immune system, but also express receptors for cytokines they produce. 

## 6. The Physiological Function of Th17

The final effect of Th17-produced cytokines, mainly IL-17A, induces the consequent production of chemokines, interleukins, and chemokines by IL17R+ cells. These factors participate to the recruitment of neutrophils to the inflammatory site and induce secretion of anti-bacterial substances by epithelial cells [7]. Importantly, Th17 is localized mainly at the mucosal level where it exerts protective activity against bacteria and fungi [69]. The primary function of Th17 cells is therefore to maintain immune control of infection at the mucosal level and in the skin [70,71]. As discussed above, the maintenance of their differentiative state over time is strictly dependent on the cytokine environment. To this end, an important role is played by low levels of IL-1β produced by macrophage cells stimulated by intestinal commensal bacteria [72]. In the skin, commensal bacteria including *S. epidermidis* contribute to Th17 cell stability [71]. The presence of several metabolites including tryptophan can also favor the differentiation state of Th17 cells [73]. At the intestinal level, Th17 cells produce IL-22 and IL-21 in addition to IL-17. IL-17 and IL-22 locally exert an antimicrobial action through the production of bactericidal proteins [74]. Either in experimental animal models or rare primary immunodeficiencies in humans, alteration in the production of interleukin 17 or its receptors as well as in the case of RORγt mutations, loss of immunological defense against *C. albicans* and *S. aureus* has been observed at the skin, nail, and genital mucosa levels [75,76]. The observation that commensal segmented filamentous bacteria (SFBs) can induce a vigorous Th17 response in the intestine appears to be of considerable importance [77]. Such bacteria have a special ability to penetrate through the mucus that protects mucous membranes and thus can resist their removal by epithelial cell turnover and digestive processes [78]. It has been proposed that this SFB property can indirectly facilitate the transformation of Th17 cells from defensive to pathogenic. Importantly, Th17 cells play their role in the mucosal defense against pathogens not only in the gut or at the skin level, but also in the lung [79,80]. Figure 2 shows the main pro-inflammatory activity of Th17 cells.

## 7. Th17 and Viral Infections

### 7.1. Animal Models 

Th17 cells play a key role in defense against viruses. In this regard, IL-17 has been shown to play a key role in H5N1 influenza virus infection. In experimental studies, mice double-knockout for the *il17* gene have shown great susceptibility to the infection and reduced survival when compared with wild-type mice [81]. Moreover, the transfer of specific Th17 cells significantly increased respiratory parameters in H5N1-infected IL-10 deficient mice [82]. Some experimental studies have shown that IL-17A is also important for herpes simplex virus (HSV) infection [83,84]. In viral myocarditis, Th17 cells were shown to mediate both immunopathology and protection. In this regard, it has been shown that, in experimental myocarditis due to Coxsackievirus B3 (CVB3) infection, there is an increase in the number of Th17 cell during the acute phase of the disease which paradoxically contributes to viral replication [85]. Downregulation of Th17 cell activity resulted in a decrease in the disease severity [86,87]. Similarly, in experimental Dengue virus (DENV) infection, high IL-17A levels are related to a bad prognosis of the disease [88].

### 7.2. Human Studies

In humans, recovery from recurrent herpes labialis is associated with an increased Th17/Treg ratio in peripheral blood [89]. On the other hand, Th17 cells have been shown to play a key role in amplifying liver inflammation during chronic hepatitis B [90,91] and C [92,93]. In congenital Zika syndrome (CZS), which can follow Zika Virus (ZIKV) infection, babies who died from brain inflammation showed a higher number of tissue-infiltrating Th17 cells as compared to controls died for unrelated causes [94]. Th17 cells and increased levels of IL-17 in peripheral blood have been reported in severe dengue infection in humans, suggesting a role for IL-17 in both the protection and pathogenesis of the disease [95,96]. In human papilloma virus (HPV) infection, specific Th17 cells infiltrate the cervical tissue in an attempt to clear the virus. However, if neoplastic transformation occurs following infection by oncogenic strains of HPV and uterine cervical cancer (UCC) develops, the pro-inflammatory activity of Th17 may lead to tumor progression by promoting angiogenesis and metastatic spread due to the destruction of the extracellular matrix [97,98]. In human immunodeficiency virus (HIV) infection, Th17 cells represent a susceptible target of the virus. Their permissiveness to infection is higher than that of Th1 and Th2 cells, as evidenced by the high level of viral DNA capable of integrating into the nucleus of Th17 cells [99]. In addition to easily entering Th17 cells due to the presence of the numerous membrane co-receptors sa4b7, CCr5, and CXCR4 [100], HIV increases in the late stages of infection the phosphorylation and expression of the target of rapamycin complex (mTOR), with a consequent increase in its replication in Th17 cells [101]. This eventually leads to an early depletion of TH 17 cells both in the peripheral blood and at the mucosal sites [102,103,104]. Interestingly, the portion of mucosal TH 17 sites is not rapidly restored following antiviral therapy, leading to severe impairment of intestinal barrier function [102,105]. The facilitated penetration of bacteria and fungi through the mucosa induces systemic immune activation and tissue inflammation, further promoting HIV replication [106,107]. Infected Th17 cells are also a critical cell compartment of the HIV reservoir. It has been reported that although Th17 cells represent only 6.2% of all CD4+ T cells, they account for approximately 18% of all HIV-infected T cells [104]. 

Recently, the role of Th17 cells has been investigated in Coronavirus Disease 2019 (COVID-19). Th17 cells have been identified in the lung of patients with COVID-19 even after viral clearance. These cells can interact with both alveolar macrophages and CD8+ cytotoxic T cells, which, following activation, participate in the “cytokine storm” responsible of the severe form of the disease [108]. Therefore, it has been proposed to block the activity of Th17 cells with anti-interleukin-17 drugs for the therapy of severe COVID-19 [109,110]. Subsequent studies have further confirmed the role of Th17 cells in the immunopathogenesis of organ damage associated with severe COVID-19 [111,112]. 

## 8. Th17 Cells and Bacterial Infections

### 8.1. Animal Models

An important role in the activation of both innate and adaptive immunity is played by commensal bacteria present in the gut [113]. In this regard, it is known that germ-free adult mice lack Th17 cells at the gut level [114]. This shows that bacteria play a key role in the generation of this T-cell subpopulation [115]. An important role in the maintenance of the Th17 response by bacteria is played by so-called superantigens. It has been shown, as mentioned earlier, that SFBs play a key role in shaping Th17 cells responses. Although the process by which these bacteria can promote the generation of Th17 cells in the intestine has not been fully elucidated, several studies have shown how SFBs can induce genes coding for serum amyloid A (SAA) and the dual oxidase 2 (Duox2) enzyme by epithelial cells [78]. SSA has been shown to be able to sensitize splenic dendritic cells to produce IL-1β. SFBs are also able to induce the generation of Th17 cells through the induction of TGF-β and IL-6. On the other hand, experiments regarding the depletion of IL-17, IL-17R, and RORγt were strictly related with SFB overgrowth [116,117]. This phenomenon was explained by the fact that these bacteria are sensitive to the action of α-defensin produced by epithelial cells in response to IL-17 [116]. Stabilization of the Th17 phenotype in the intestine is further promoted by IL-23 produced by dendritic cells of the lamina propria after stimulation by SFBs [78,118]. On the other hand, the presence of bacteria that can penetrate the mucosa and persist within the intestine can promote self-reactive Th17 cell responses [78,119]. As already pointed out, Th17 cells play an important role in mucosal defense against extracellular bacteria. IL-17R-deficient mice succumb 100% following an attack by klebsiella as they cannot activate a defensive response by neutrophils [120]. Th17 cells also play a role against intracellular bacteria. In the case of *M. tuberculosis*, a Th17 response was found to be necessary for a defense against the primary infection [121,122]. Indeed, in the early stages, Th17 cells facilitated tissue recruitment of neutrophils, macrophages, and Th1 cells to the areas of mycobacterium penetration. Moreover, Th17 cells were found to be responsible for the development of protective Th1 cells in the later stages of infection [123]. It has been reported, however, that Th17 response can be pathological rather than protective since a correlation with pulmonary tuberculosis progression and distant dissemination of infection has been observed in experimental models [121,124]. With regard to respiratory allergic diseases, studies in animal models have shown that airway inflammation is attenuated in allergic mice double-knockout for the *il17a* gene. TNF-α blockade has been shown to reduce the production of both IFN-γ and IL-17, suggesting that *M. catarrhalis* infection is dependent on both cytokines [125]. In addition, it has been shown how *H. influenzae* infection can promote a Th17 response with the consequent recruitment of neutrophils to the mucous membranes of the respiratory tract [126], possibly leading to airway remodeling and an increase in mucin secretion during OVA-induced allergy sensitization. Although there is evidence that *M. pneumoniae* infection may also be a trigger for asthma, the exact mechanism is still unclear. However, some studies have found that IL-17A in the lungs is markedly elevated in mice after infection with this pathogen. Experiments in vitro have found that the levels of IL-17A in the culture medium increased with increasing *M. pneumoniae* concentration [127]. Therefore, there is evidence that the IL-23/Th17 axis can be critical for asthma onset in the course of *M. pneumoniae* infection. In support of this hypothesis, mice infected with *M. pneumoniae* exhibited increased IL-23 secretion by alveolar macrophages, and increased levels of IL-17A and IL-17F as well. After the pharmacological neutralization of IL-23, the production of IL-17A and IL-17F was blocked resulting in decreased neutrophil recruitment in the lungs [128]. It was also hypothesized that *M. pneumoniae* infection could activate IL-6/STAT3 promoting Th17 cell differentiation and cytokine secretion with the consequent development of asthma [129]. Finally, it has been shown that mice lacking IL-17R are more sensitive to P. gingivalis-induced bone loss, demonstrating a protective role of IL-17 in bone homeostasis [130].

### 8.2. Human Studies

It has been observed that the superantigen Toxic Shock Syndrome Toxin-1 (TSST-1) produced by *Staphylococcus aureus* can stimulate autoreactive T cells in an antigen-independent manner [131]. Induction of interleukin 17 production by Th17 cells has been observed during *S. aureus* infection in response to enterotoxin B (SEB). In this regard, it is interesting to note that Toll-like receptors (TLRs) receptors expressed by cells of the innate immune system play an important role in defense against *S. aureus* after activation by pathogen-associated molecular patterns (PAMPs). The stimulation of TLR2 promotes a Th17-mediated reaction with possible onset of granulomatosis with polyangiitis [132]. Reduced Firmicutes:Bacteroidetes ratio in the microbiota has been associated with increased Th17 cell activity in patients with systemic lupus erythematosus (SLE) [133]. Gut dysbiosis has been also associated with other autoimmune diseases, including inflammatory bowel disease [134], multiple sclerosis [135], rheumatoid arthritis [136], and myasthenia gravis [137]. A particularly important role is played by Th17 cells in the genesis of chronic inflammatory bowel disease (IBD). In such patients, the percentage of Th17 cells is significantly increased in both blood and intestinal mucosa. However, in the case of abnormal activation of Th17 cells, increased levels of IL-17, IL-21 21, and IL-23 are detected with possible induction of IBDs [138]. Under such conditions, increased stimulation of fibroblasts and epithelial damage through the production of metalloproteinase were observed [139]. The close correlation between gut bacterial flora and the protective or immuno-pathogenic function of Th17 cells led to the hypothesis that manipulation of the gut microbiota could have a therapeutic effect on IBDs [140]. It has been shown in this regard that sterilization of the microbiota by antibiotic therapy can significantly reduce the differentiation of Th17 cells, and this can be restored after fecal transplantation [141]. Some studies have also shown that the use of probiotics that restore the homeostasis of intestinal bacterial flora can inhibit the proinflammatory effect of Th17 cells by improving the IBD progression [142,143]. In another study, the use of polysaccharide A derived from Bacteroides fragilis resulted in the amelioration of such inflammatory diseases by suppressing the pro-inflammatory response of intestinal Th17 cells through the stimulation of TLR2 [144]. Other studies have shown how gut barrier dysfunction contributes to worsening disease in patients with advanced cirrhosis [145,146]. Moreover, alterations in intestinal bacterial flora with increased Th17 activity were found to play an important role in the pathogenesis of inflammatory rheumatic diseases, including rheumatoid arthritis [147] and HLA-B27-associated spondylarthritis [148]. 

Th17 cells play also an important role in the genesis of many respiratory diseases. In particular, these T cells have been implicated in the occurrence of asthma. Several studies have shown how children who were infected perinatally with pathogens, such as *H. influenzae*, *S. pneumoniae*, and *M. catarrhalis*, have an increased risk of developing bronchial asthma [149]. 

*S. pneumoniae* is a pathogen capable of causing community-acquired pneumonia. Perinatal infection with this pathogen is closely related to the subsequent development of asthma. Several studies show that *S. Pneumoniae* infection is associated with increased numbers of Th17 cells in the airways with concomitant overproduction of interleukin 17A [150,151]. *M. catarrhalis* is another mucosal pathogen that causes respiratory illness in children. Infection by this pathogen has been found to increase the risk of asthma in both infants and adults. It has been also observed that IL-17A levels are significantly increased in the lungs of dust mite-allergic mice infected with *M. catarrhalis*.

*M. pneumoniae* is an airway pathogen that mainly adheres to the surface of the respiratory tract. *M. pneumoniae* infection can cause chronic inflammation of the lower airways by impairing ciliary clearance and increasing mucus secretion. Interestingly, it has been shown that a significant percentage of subjects with refractory asthma are infected with this pathogen [152,153]. Th17 cells were also found at the level of gingival tissues [130]. Although IL-17 may have protective functions in the oral cavity, several studies indicate that its excessive production is associated with periodontitis [154]. In this regard, it has been shown that IL-17 is responsible for pro-inflammatory activities by inducing cytokines via target cells and recruiting inflammatory cells such as neutrophils in the oral cavity. In addition, IL-17 facilitates the access of these cells to tissues through the regulation of chemokine ligand expression and granulocyte-macrophage colony-stimulating factors (GM-CSF). Upregulation of IL-17 can lead to the overactivation and mobilization of macrophages eventually leading to oral tissue damage [155]. In addition, IL-17 acts synergistically with other inflammatory cytokines to increase chemokine production by human gingival fibroblasts. This stimulates further recruitment of Th17 cells and, consequently, IL-17 production in inflamed periodontal tissues. A significant increase in IL-17 levels has been found not only in the bone and gingiva of periodontitis patients, but also in serum [156]. Expression levels of the RORC encoding gene have been found to be increased in patients with periodontitis [157]. 

## 9. Th17 Cells and Fungal Infections

### 9.1. Animal Models

Th17 cells have been proven to be critical in defense against fungi and in particular against *Candida albicans*, a mucocutaneous fungal pathobiont [158]. Animals made deficient for the expression of IL-17, IL-17R, or RORγt proved particularly susceptible to *Candida albicans* infection [75]. However, repeated fungal infections have been shown to activate Th17 cells and induce immunopathology. With regard to other fungal infections, it has been shown that *A. alternata* can induce a Th17 response in the lungs, activating neutrophils by β-glucan, which is able in turn to induce Th17 cells by binding Dectin-1. This promotes the secretion of several cytokines with consequent amplification of the inflammatory response [159,160,161]. 

### 9.2. Human Studies

Although fungus-specific Th17 cells are present in peripheral blood in almost all healthy subjects [162], it has been suggested that the role of fungi in inducing immunopathology is significantly underestimated probably due to a lack of adequate diagnostic tools [163,164]. *C. albicans* in particular is the most potent pathogen in humans capable of stimulating a Th17 response even more than *S. aureus* [51,165]. Such a response is induced by the peptide toxin Candidalysin. This protein not only protects against fungal invasion, but is also capable of damaging the intestinal epithelium [166,167]. Moreover, it has recently been demonstrated that A. fumigatus-specific T cells re-stimulated with *C. albicans* could acquire a Th17 phenotype, although these cells have been proven not to be essential for defense against this fungus [158,162]. It was also observed that patients with lung diseases are more easily colonized by *A. fumigatus* [168,169] and that such patients show more severe forms of pulmonary disease when the lung is infiltrated by Th17 cells [170,171,172]. This demonstrates the immunopathological activity of this T-cell subset in the lung. Therefore, it can be speculated that Th17 cells present at the mucosal level where they are protective against *C. albicans* can generate pathogenic A. fumigatus-specific through a heterologous cross-reactive immunologic response [173]. These findings also have therapeutic implications. Indeed, the overuse of antibiotic therapy results in increased intestinal colonization by *C. albicans* [174,175]. This has been correlated with pathology at sites far from the gut including the lung [176]. Therefore, preservation of the microbiota may be a key factor in counteracting the immunopathologic remote effects mediated by cross-reactive Th17 cells. Figure 3 represents the dual effect of the interaction of Th17 cells with pathogens.

## 10. Conclusions

Several factors can transform Th17 cells from an effective defensive arm against infectious pathogens into harmful self-reactive effectors. The pathogens themselves play a key role in this conversion. It is quite clear that gut commensal bacteria in the microbiota play a key role. However, as discussed in the previous sections, viruses and fungi can also induce Th17 cells to acquire a pathogenic phenotype. Th17 cells rendered pathogenic after interaction with various microbial agents are involved in the genesis of several autoimmune rheumatic diseases, such as rheumatoid arthritis, systemic lupus erythematosus, myasthenia gravis, and vasculitis, but this list is not certainly exhaustive. They also play a crucial role in psoriasis. An important role of Th17s in promoting the genesis of various neoplasms is also beginning to emerge [177]. In the case of HIV infection, moreover, Th17 cells perform a peculiar function by constituting a long-term HIV reservoir within the body. Quite recently Th17 cells are critical in the genesis of the “cytokine storm” characteristic of severe COVID-19. Luckily, after the identification of the pathogenetic role of Th17 cells, powerful therapeutic tools, such as monoclonal antibodies blocking the IL 23-IL 17 axis, have been developed. Future studies are needed to further elucidate the mechanisms by which Th17 cells can cause tissue damage following interaction with infectious agents and how to therapeutically modulate the activity of this cell subpopulation to be beneficial but not harmful to the host.

## Figures and Tables

**Figure 1 pathogens-11-01547-f001:**
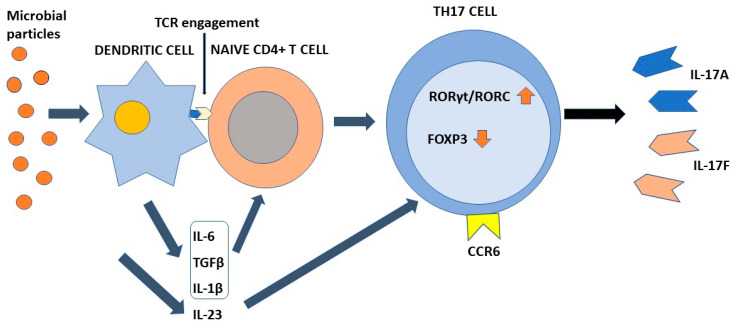
**The process of Th17 cell differentiation.** Dendritic cells after activation by microbial components activate CD4+ naive T cells in an antigen-dependent manner. In the presence of the appropriate cytokine environment, such cells acquire a Th17 phenotype by upregulating RORγt/RORC and downregulating Foxp3. IL-23 is required to stabilize their phenotype. Th17 cells are characterized by the ability to synthesize and secrete IL-17A and IL-17F in addition to other soluble factors. IL-6, TGFβ, and IL-1β contribute to the differentiation of naïve T cells into Th17 cells. IL-23 is required to stabilize their phenotype. Chemokine receptor CCR6 directs Th17 cells to barrier tissues.

**Figure 2 pathogens-11-01547-f002:**
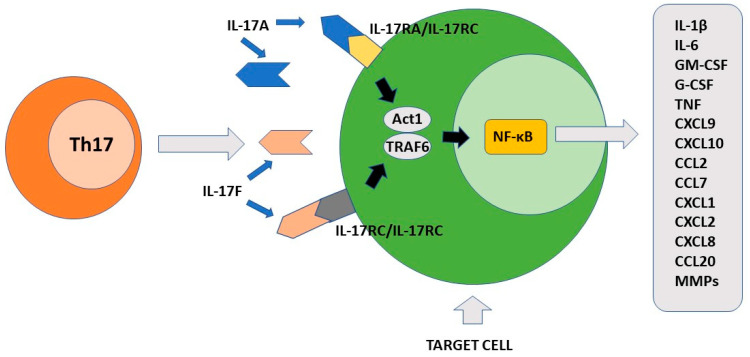
**The pro-inflammatory function of Th17 cells.** IL-17A and IL-17F produced by Th17 cells recognize different cellular targets that express their specific receptors. After activation, the signal transduction is triggered with formation of the ACT1/TRAF6 complex. The final event of this process includes activation of transcription factor NF-κB. Different pro-inflammatory factors are synthesized by the stimulated cells including cytokines, chemokines, growth factors and matrix metalloproteinases. IL-17A/IL-17F target cells include keratinocytes, fibroblasts, osteoblasts, epithelial cells, endothelial cells and macrophages.

**Figure 3 pathogens-11-01547-f003:**
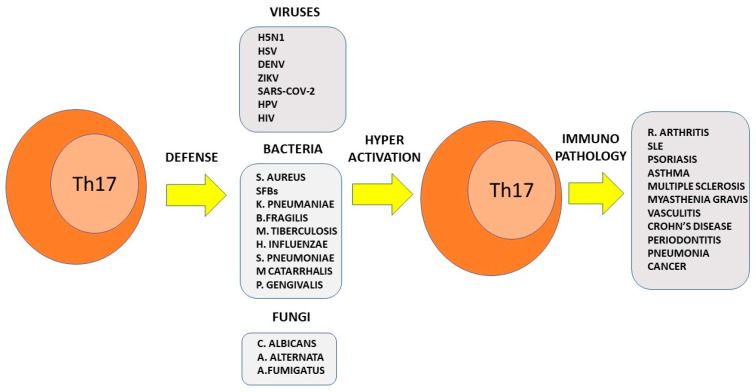
The dual role of Th17s after encountering pathogens. Th17 cells provide mucosal defense against viral, bacterial, and fungal agents. However, the same pathogens can induce altered activation of Th17 cells, which may in turn become responsible for immunopathology causing disease onset. Evidence that the Th17 response to pathogens can mediate immunopathology comes from findings in humans, animal models, or both.

## Data Availability

Not applicable.

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
