# Peer review of "The Double Game Played by Th17 Cells in Infection: Host Defense and Immunopathology"

_pathogens, 2022, doi:10.3390/pathogens11121547_

Round 1

Reviewer 1 Report

It is a well-written review that covered the basic aspects of TH17 cells.

Author Response

Responses to reviewer no. 1

  1. Query: English language and style are fine/minor spell check required. Answer: English has been revised and the style has been improved

Reviewer 2 Report

Marino Paroli et al. reviewed the role of T cells producing interleukin 17 known as Th17 cells in defense against viruses, bacteria, and fungi and their undesirable contribution to pathology. The manuscript is well written although it needs to be organized in some sections.

Minor improvements needed:   

1.      In my opinion, the authors need to include references in the introduction.

2.      The authors need to make a clear distinction when referring to human and animal model studies. The authors can group the findings from animal models followed by human studies.

3.      The authors referred to only one study of Th17 in dengue from a mouse model but there are studies of IL-17 in serum and Th17 cells in people infected with the dengue virus that need to be included (PMID 23274801, 33322218).

4.      The authors may consider approaching the order of the viral infections. For example, Th17 cells in HPV, and HIV followed by Th17 in COVID-19.

5.      Section 8, first paragraph. There is a typo – “Th717”

6.      Figure 1. There are eight circles in orange and one in red; is that telling us something? If so, please describe it in the legend. In addition, there is a box including IL-6, TGF-β, and IL-1β while IL-23 is separate, so I think it would be great if you can describe the meaning in the legend. It may be important to add the CCR6 to the Th17 cells in the figure.   

7.      Figure 2. The authors may consider including some examples of the target cells in the figure legend.

8.      Figure 3. It may be important to mention that the evidence of Th17 to different pathogens comes from findings in humans, animal models, or both.

Author Response

Responses to reviewer no. 2

  1. Query: In my opinion, the authors need to include references in the introduction. Answer: References have been added in the introduction section.
  2. Query: The authors need to make a clear distinction when referring to human and animal model studies. The authors can group the findings from animal models followed by human studies. Answer: The chapters on pathogen response have been separated into two subheadings: animal models and human studies.
  3. Query: The authors referred to only one study of Th17 in dengue from a mouse model but there are studies of IL-17 in serum and Th17 cells in people infected with the dengue virus that need to be included (PMID 23274801, 33322218). Answer: The two references were added with a brief commentary
  4. Query: The authors may consider approaching the order of the viral infections. For example, Th17 cells in HPV, and HIV followed by Th17 in COVID-19. Answer: The order of the discussion of the different viral infections has been changed as suggested
  5. Query: Section 8, first paragraph. There is a typo – “Th717” Answer: The typo has been corrected
  6. Query: Figure 1. There are eight circles in orange and one in red; is that telling us something? If so, please describe it in the legend. In addition, there is a box including IL-6, TGF-β, and IL-1β while IL-23 is separate, so I think it would be great if you can describe the meaning in the legend. It may be important to add the CCR6 to the Th17 cells in the figure. Answer: The color of the red circle has been changed making it the same as the others. A representation of the CCR6 receptor has been added to the Th17 cell figure with a brief comment in the legend. The role of IL-23 and the cytokines in the box has also been commented in the legend
  7. Query: Figure 2. The authors may consider including some examples of the target cells in the figure legend. Answer: A list of IL-17A and IL-17 F target cells has been added to the legend
  8. Query: Figure 3. It may be important to mention that the evidence of Th17 to different pathogens comes from findings in humans, animal models, or both. Answer: Evidence that the data shown in the figure are derived from findings in humans, animal models, or both was specified in the legend